

# Your blush gives you away: detecting hidden mental states with remote photoplethysmography and thermal imaging

Ivan Liu[1,2], Fangyuan Liu[2], Qi Zhong[1], Fei Ma[3] and Shiguang Ni[4]

[1] Faculty of Psychology, Beijing Normal University, Beijing, China
[2] Department of Psychology, Faculty of Arts and Sciences, Beijing Normal University at Zhuhai, Zhuhai, Guangdong, China
[3] Guangdong Laboratory of Artificial Intelligence and Digital Economy (SZ), Shenzhen, Guangdong, China
[4] Shenzhen International Graduate School, Tsinghua University, Shenzhen, Guangdong, China

Corresponding author
Shiguang Ni,
ni.shiguang@sz.tsinghua.edu.cn

## ABSTRACT

Multimodal emotion recognition techniques are increasingly essential for assessing mental states. Image-based methods, however, tend to focus predominantly on overt visual cues and often overlook subtler mental state changes. Psychophysiological research has demonstrated that heart rate (HR) and skin temperature are effective in detecting autonomic nervous system (ANS) activities, thereby revealing these subtle changes. However, traditional HR tools are generally more costly and less portable, while skin temperature analysis usually necessitates extensive manual processing. Advances in remote photoplethysmography (r-PPG) and automatic thermal region of interest (ROI) detection algorithms have been developed to address these issues, yet their accuracy in practical applications remains limited. This study aims to bridge this gap by integrating r-PPG with thermal imaging to enhance prediction performance. Ninety participants completed a 20-min questionnaire to induce cognitive stress, followed by watching a film aimed at eliciting moral elevation. The results demonstrate that the combination of r-PPG and thermal imaging effectively detects emotional shifts. Using r-PPG alone, the prediction accuracy was 77% for cognitive stress and 61% for moral elevation, as determined by a support vector machine (SVM). Thermal imaging alone achieved 79% accuracy for cognitive stress and 78% for moral elevation, utilizing a random forest (RF) algorithm. An early fusion strategy of these modalities significantly improved accuracies, achieving 87% for cognitive stress and 83% for moral elevation using RF. Further analysis, which utilized statistical metrics and explainable machine learning methods including SHapley Additive exPlanations (SHAP), highlighted key features and clarified the relationship between cardiac responses and facial temperature variations. Notably, it was observed that cardiovascular features derived from r-PPG models had a more pronounced influence in data fusion, despite thermal imaging's higher predictive accuracy in unimodal analysis.

## INTRODUCTION

Over the past two decades, the use of multimodal emotion recognition techniques (MMER) in mental state assessment has gained increasing traction, offering profound insights in fields as varied as marketing, education, and mental health (*Bahreini, Nadolski & Westera, 2016*; *Soleymani, Pantic & Pun, 2011*). MMER primarily utilizes image-based methodologies, analyzing facial expressions, body movements, gestures, and eye movements to assess psychological states. These methods, leveraging only camera technology, are cost-effective and non-intrusive, making them suitable for a wide range of applications. Additionally, they resonate with human visual perception, producing results that are intuitively understandable and easily explainable. However, image-based MMER primarily detects basic emotions that significantly alter appearance or behavior, such as anger, surprise, disgust, enjoyment, fear, and sadness (*Ekman, 1992*). These methods often depend on obvious visual cues, overlooking the subtler nuances of emotional states.

Psychophysiological research, rooted in neuroscience, shows that physiological markers, namely heart rate (HR) and skin temperature, serve as reliable indicators of changes in mental states. These changes are reflected in autonomic balance alterations, characterized by either activation of the sympathetic nervous system (SNS) or suppression of the parasympathetic nervous system (PNS). Such dynamics lead to an increase in HR as a response to perceived threats, whereas a decrease in SNS activity along with an increase in PNS function correlates with HR reduction during relaxation phases. Given the autonomic nervous system (ANS), which encompasses both SNS and PNS, is regulated by the prefrontal cortex, and considering that mental exertions significantly tax cognitive resources and affect prefrontal cortical functions, HR fluctuations have been linked to various cognitive and affective processes. These include stress response modulation (*Cho, Julier & Bianchi-Berthouze, 2019*), sustained attention (*Widjaja et al., 2015*), and emotional responses to moral beauty (*Piper, Saslow & Saturn, 2015*).

The activation of ANS plays a crucial role in thermoregulatory responses, with SNS activation in response to perceived threats leading to peripheral vasoconstriction. This reaction causes a reduction in cutaneous blood flow and, consequently, a decrease in surface body temperature (*Kistler, Mariauzouls & von Berlepsch, 1998*). However, cutaneous temperature changes are not solely dependent on these factors; they are also influenced by sudomotor activity (sweating), muscular contractions, and lacrimation. Research has demonstrated that emotions linked to sympathetic arousal, such as fear and anxiety, lead to a reduction in dermal temperature, particularly noticeable in the peripheral extremities and facial regions like the cheeks and nasal tips. The nasal tips, in particular, tend to exhibit a more pronounced response to stress (*Engert et al., 2014*). On the contrary, fear and anxiety may also increase muscular activity in the forehead and periorbital

regions, resulting in a temperature increase in these areas (*Levine, Pavlidis & Cooper, 2001*; *Pavlidis & Levine, 2002*; *Vinkers et al., 2013*). Furthermore, a positive correlation exists between sustained cognitive engagement and an increase in forehead temperature (*Bando, Oiwa & Nozawa, 2017*).

Recent developments in psychophysiological research have expanded to include the thermal effects of various emotional states. *Salazar-López et al. (2015)* noted that nasal temperature typically decreases in response to negative valence stimuli, but it also increases with positive emotions and arousal. Interestingly, these changes in nasal temperature positively correlate with participants' empathy scores and emotions like love. In a study involving fifteen three-year-olds, *Ioannou et al. (2013)* observed that sympathetic arousal caused by toy malfunctions led to a significant drop in nasal temperature. When the children were comforted, nasal temperature increased, indicative of parasympathetic activation and suggesting either distress alleviation or overcompensation. Additionally, *Ioannou et al. (2016)* found that sympathetic crying induced by sad films in female subjects resulted in increased temperatures in the forehead, periorbital region, cheeks, and chin. Conversely, the maxillary area showed a decrease in temperature, attributed to emotional sweating.

Cardiovascular data and thermal imaging are instrumental in uncovering concealed emotions, a key aspect of psychological analysis and various applications. However, their widespread practical application faces significant challenges. A primary obstacle is the costly and intrusive nature of heart rate detection tools such as electrocardiograms (ECG) and pulse oximeters. These devices also suffer from a lack of portability. Despite technological advancements yielding more portable commercial devices, these non-image-based methods are still not user-friendly. They necessitate the purchase of additional equipment and the need for users to carry these devices consistently. However, while thermal imaging offers a less intrusive alternative, the requirement for manual data cleaning and processing, particularly in identifying regions of interest (ROI), is a tedious and time-consuming task. This increased labor intensity and associated costs dampen enthusiasm for both research and practical usage, thus hindering the exploration of their full potential, such as in continuous monitoring scenarios.

The recent advancements in image-based heart rate detection and automatic thermal ROI detection methods shed light on the aforementioned problem. Photoplethysmography (PPG), an optical method, detects blood volume changes beneath the skin due to heartbeats (*Elgendi, 2012*). As hemoglobin's light absorption differs, blood volume changes are identified by observing the reflected light intensities. Traditionally, the contact PPG signal is obtained by employing finger oximeters with LED light (*Takano & Ohta, 2007*). Remote PPG (r-PPG), on the other hand, detects heartbeats by recording videos of faces and converting the facial skin color changes into waveforms. The main advantage of r-PPG is its capacity for non-invasive, continuous vital sign monitoring. However, the difficulty in obtaining high-quality signals curtails its acceptance both in research and in practice. While many studies have managed to produce sufficiently accurate average HR measurements—due to the robustness in calculating the average HR when signal quality is low—this metric offers limited insight into autonomic nervous system (ANS) activity,

making it less pertinent for psychological studies (*Yu, Li & Zhao, 2019*). Conversely, while heart rate variability (HRV) offers a richer source of psychophysiological information (*Liu, Ni & Peng, 2020b*), it is more susceptible to slight alterations in environmental lighting and facial movements.

Encouragingly, recent advances in signal processing and machine learning have markedly improved the precision of HRV metrics derived from r-PPG data. *Huang & Dung (2016)* employed a smartphone camera and utilized continuous wavelet transform to mitigate noise, which reduced the mean absolute error (MAE) for r-PPG with the referencing data from 20 to 2. Similarly, *Qiao et al. (2021)* harnessed green light from the cheek and nose regions and achieved an MAE of 24.33 ± 28.66 using Welch's method. *Yu, Li & Zhao (2019)* capitalized on Spatio-Temporal Networks to curtail errors, boosting the correlation between r-PPG and ECG to 0.766 on the OBF dataset. An early exploration by *McDuff, Gontarek & Picard (2014)* into emotion recognition using r-PPG yielded a 0.93 correlation for high-frequency HRV. Employing a digital camera with five color bands, they successfully categorized three emotional states with an 85% accuracy rate. Nevertheless, the broader application of r-PPG in producing HRV metrics for nuanced mental state detection remains somewhat nascent.

Manual tracking of ROIs in thermal imaging processing is tedious, especially for larger datasets or prolonged monitoring. Consequently, many prior psychological studies opted for a simplified approach, manually analyzing temperature shifts before and after a stimulus was applied (*Ioannou et al., 2016*). Many early ventures into thermal imaging, on the other hand, navigated around obstacles by asking participants to stay still (*Pavlidis & Levine, 2002*), thereby limiting its applicability in real-world scenarios. Notably, advancements in machine learning-based ROI detection enable consistent temperature tracking (*Joshi, Bianchi-Berthouze & Cho, 2022*), even with slight head movement (*Cho et al., 2019*; *Kuzdeuov et al., 2022*). Consequently, *Cruz-Albarran et al. (2017)* identified basic emotions such as joy, disgust, anger, fear, and sadness, achieving an impressive accuracy of 89.9%. Similarly, *Goulart et al. (2019)* crafted a model that delivered prediction accuracies of 89.88% for disgust, 88.22% for happiness, 86.93% for surprise, 86.57% for fear, and 74.70% for sadness among children aged between seven and eleven.

Recent developments in r-PPG and thermal imaging show promise, yet there is a critical need to further enhance their accuracy. The success of automatic ROI detection, pivotal in the initial data processing stages for both r-PPG and thermal imaging, is greatly influenced by data quality. This quality hinges on various factors, such as individual movements, facial obstructions (like glasses or hair), camera angles, and environmental lighting and temperature changes. Although current signal processing methods hold potential, they typically produce satisfactory results in laboratory settings with controlled conditions. This underscores the urgent need for more advanced signal processing techniques or machine learning algorithms, essential for improving the reliability and accuracy of r-PPG and thermal imaging, particularly in real-world, uncontrolled environments.

While researchers in the r-PPG and thermography fields struggle to mitigate the inherent low signal quality issue, they often overlook the potential of combining both methods to improve prediction accuracy further. Single-source physiological data often

lacks accuracy (*Dino et al., 2020*). In contrast, data analysis across different modalities can complement each other, reducing randomness and enhancing robustness. As a result, MMER studies demonstrate superior performance compared to their single-modality counterparts (*Morency, Mihalcea & Doshi, 2011*; *Sebe et al., 2006*; *Wang et al., 2010*; *Zhao et al., 2021*). Besides, both remote PPG and thermal imaging methods can collect physiological signals non-intrusively over long periods, and despite differing in their physiological mechanism, their similar data collection, environmental, and equipment requirements make them ideal for integration. As of today, only a few pioneering studies have explored the combination of HR with thermal imaging (*Cho, Julier & Bianchi-Berthouze, 2019*). However, to this best knowledge of the authors, there have been no attempts to extend the literature to include the use of r-PPG, which is more suitable for use with thermal imaging as a remote ANS detection tool.

This research aims to bridge a significant gap in current literature by exploring the integration of r-PPG and thermography to improve the accuracy of predicting changes in psychological states. The study focuses on comparing early and late data fusion strategies and employing two prevalent machine learning models: support vector machine (SVM) and random forest (RF). The goal is to ascertain how these two modalities can be effectively integrated to develop an enhanced predictive model.

A critical aspect of this research involves identifying key features within the predictive model to elucidate the complex relationship between cardiovascular features, facial expressions, and psychological states. Conducting a comprehensive examination of these features is essential for optimizing model performance and facilitating more accurate adjustments and interpretations (*Du, Liu & Hu, 2019*; *Murdoch et al., 2019*). Such transparency is particularly vital in areas such as healthcare where the opacity of machine learning models presents interpretive challenges and restricts their practical application potential (*London, 2019*; *Tonekaboni et al., 2019*; *Vellido, 2020*). Moreover, an in-depth exploration of these features yields greater insights into psychophysiology and extend the understanding to the physiological responses of various mental state changes.

To meet these goals, the study conducts laboratory experiments to collect data from participants experiencing cognitive stress and moral elevation. These conditions represent the spectrum of negative and positive emotional state changes that lead to ANS-related variations in heart rate and skin temperature. Cognitive stress, a common precursor to psychological issues, is known to elicit various physiological responses, including changes in HRV and skin temperature (*Cho, Julier & Bianchi-Berthouze, 2019*). On the other hand, moral elevation, defined by *Haidt (2000)* as a positive emotional response to witnessing acts of kindness and compassion, fosters a sense of warmth and promotes prosocial behavior (*Haidt, 2003*). It intensifies the desire to help others (*Han et al., 2015*) and enriches life's purpose understanding (*Oliver, Hartmann & Woolley, 2012*). Previous studies have linked moral elevation to ANS activity (*Silvers & Haidt, 2008*) and HRV (*Piper, Saslow & Saturn, 2015*). Additionally, moral elevation can trigger physical sensations like chest expansion, exhilaration, tearing up, goosebumps, and a warming sensation in the chest area (*Algoe & Haidt, 2009*), potentially influencing facial temperature changes.

## MATERIALS AND METHODS

Portions of this text were previously published as part of a preprint (https://doi.org/10.48550/arXiv.2401.09145).

### Participants and experiment procedures

This research forms a part of a larger study focused on the feasibility of MMER. The research protocol was approved by the ethics committee of the faculty of psychology at Beijing Normal University (No. 202203070037), and written informed consent was received from all participants. For their involvement, each participant received a compensation of 150 RMB (approximately 20 USD).

The experimental procedure commenced with a briefing, after which participants were instructed to remain as still as possible while completing a 20-min questionnaire. This process, inspired by *McDuff, Gontarek & Picard (2014)*, involved collecting r-PPG and thermal imaging data to assess physiological responses to cognitive stress caused by prolonged task focus (*Tanaka, Ishii & Watanabe, 2014*). A built-in camera on a notebook (ThinkPad E310, Lenovo, Beijing, China) recorded participants' facial reactions. Concurrently, a thermal camera (One Pro, FLIR, Wilsonville, OR, USA) to their right measured facial temperatures. ECG data (AD8232 ECG module, Sichiray, Shenzhen, China) were also gathered using a custom Arduino system. While minimal movement was allowed, participants were encouraged to limit it. Out of 104 participants (21% male, average age 21.33, SD 2.45), ninety viewed a short film on firefighters' sacrifice, aimed at inducing moral elevation. This session's r-PPG and thermal imaging data were used for developing a moral elevation prediction model. Among all participants, 86 provided valid r-PPG data, and 55 yielded valid thermal imaging data. After watching, they completed *Aquino, McFerran & Laven*'s *(2011)* moral elevation scale, with a *t*-test validating the film's efficacy in eliciting moral elevation ($p < 0.001$).

### R-PPG

#### *Signal extraction*

This study analyzed all data with the Python package pyMMER (available at https://github.com/8n98324n/pyMMER), developed for this study. PyMMER integrates several publicly available open-source Python packages to help non-technically savvy researchers process and analyze multimodal data for research. For r-PPG data processing, adapting the code from the Python package pyVHR (*Boccignone et al., 2022*), pyMMER first identifies patches of ROIs using MediaPipe Face Mesh and continuously tracks them in all video frames (Fig. 1). Out of the 468 facial ROI identified in MediaPipe (Google Inc, Mountain View, CA, USA), this study deliberately used only 70 regions, excluding regions close to the edges of the face, lips, and eyes to mitigate the potential interference from spontaneous facial movements and to factor in participants who wore glasses.

PyMMER computes average color intensities for each patch across overlapping windows, producing multiple time-varying RGB signals for each temporal segment. During the signal processing phase, pyMMER utilizes the plane-orthogonal-to-skin (POS) method, as described by *Wang et al. (2016)*, to convert these signals into a pulse waveform

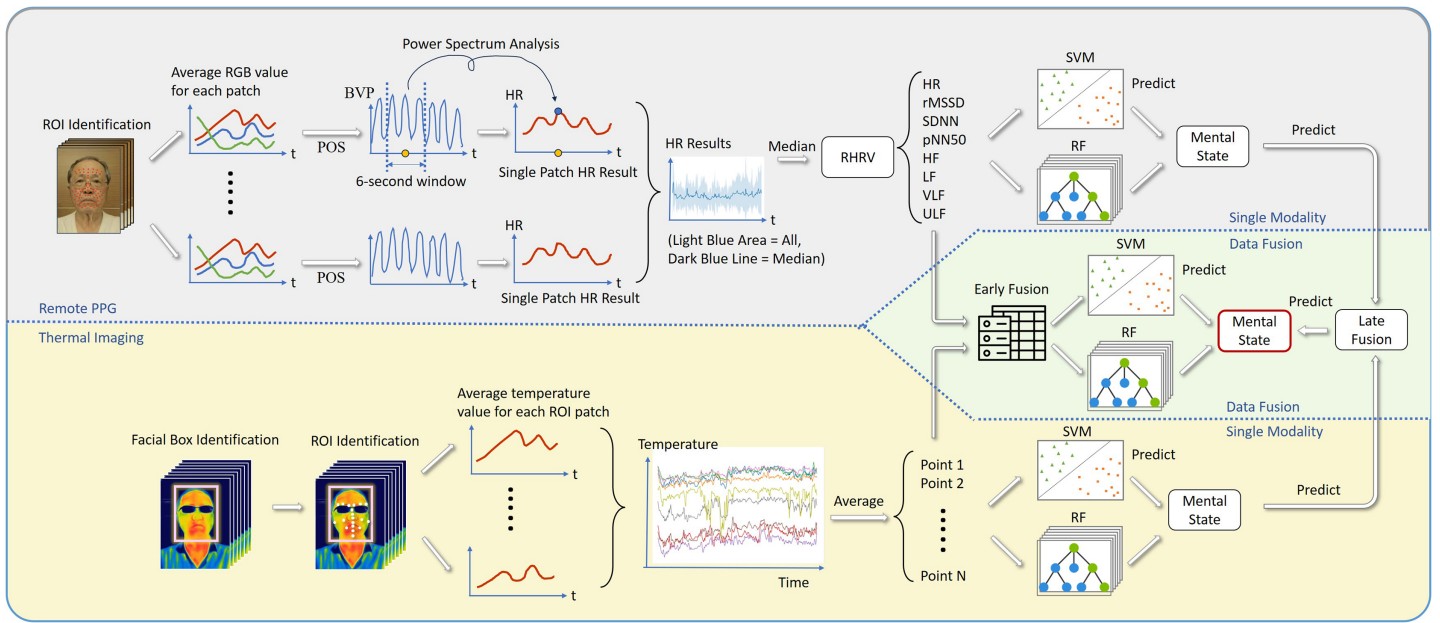

**Figure 1 The signal processing flow.**

(blood volume pulse, BVP). As highlighted by *Boccignone et al. (2022)*, this method ranks among the top performers in their study. pyMMER then segments the BVP into overlapping 6-s windows. For each window, pyMMER determines the HR (measured in beats per minute, BPM) by identifying the most significant frequency in the power spectrum of the wave, generated within the 6-s window using Fourier analysis.

### HRV data processing

After obtaining BPM data, pyMMER identifies problematic HR points that change by more than 25 beats per minute from the previous points. It then removes either the current or the previous data point that is further from the median HR of the dataset. Subsequently, pyMMER utilizes the R language package RHRV (*Martínez et al., 2017*) to calculate HRV measures. If HR or HRV exceeds a predefined threshold, the RHRV package is unable to produce HRV measures, and such data are considered outliers in this study.

There are two main types of HRV measures: time-domain and frequency-domain. The time-domain indices of HRV quantify variability in the beat-to-beat interval (BBI). This study included three commonly used time-domain measures for comparison: Root Mean Square of Successive Differences (rMSSD), Standard Deviation of NN Intervals (SDNN), and the percentage of NN interval changes larger than 50 ms (pNN50). The frequency-domain components of HRV consist of four frequency bands: high frequency (HF), low frequency (LF), very low frequency (VLF), and ultra-low frequency (ULF). Given that this study only recorded 5-min videos and used 2-min segments for analysis, the ULF and VLF bands do not apply (*Shaffer & Ginsberg, 2017*). The HF and LF values were then log-

transformed because they were not distributed normally (*Chalmers et al., 2016*; *Laborde, Mosley & Thayer, 2017*).

### ECG referencing

To ascertain the accuracy of r-PPG, this study compared HRV measurements from the r-PPG against a reference ECG. This study employed the Python package py-ecg-detectors (*Howell & Porr, 2023*) to transform raw ECG signals into heart rate data, utilizing the QRS detection algorithm proposed by *Elgendi, Jonkman & De Boer (2010)*. This study collected 383 samples with both valid r-PPG and valid ECG data from all participants. Data from three participants were then manually excluded because the data collected from r-PPG and ECG were significantly different, possibly due to collection error. In order to achieve acceptable signal quality, previous studies have argued the importance of using a quality index to filter out potentially corrupted data (*Liu, Ni & Peng, 2020a*). Since there are no established quality criteria for remote PPG results generated from multiple ROIs, this study suggested using the MAE of the HR over HR (MAE/HR) obtained from all ROIs as the quality index. This study then compared the correlation coefficient and *p*-value of the HR and HRV measures.

## Thermal imaging
### Feature extraction

For thermal imaging processing, 'pyMMER' integrates the open-source Python code provided by *Abdrakhmanova et al. (2021)* and *Kuzdeuov et al. (2022)* (https://GitHub.com/IS2AI/thermal-facial-landmarks-detection). Their project trained a ROI detection model based on the Histogram of Oriented Gradients (HOG) and SVM methods. The dataset contained 2,556 facial thermal images of 142 subjects with manually annotated face bounding boxes and 54 facial landmarks. In cases where the trained model could not identify boxes of faces, these instances were also classified as outliers in this study. Besides, as many participants wore glasses, this study avoided the orbital region, focusing exclusively on twenty-two ROIs (Table 1).

## RESULTS
### Equipment accuracy analysis
### ECG referencing

The ECG referencing analysis showed that the correlation coefficients for HR and all time-domain HRV measures increased almost monotonously when MAE/HR decreased, suggesting that MAE/HR was a robust and effective quality criterion (Table 2). Based on these results, this study selected MAE/HR = 0.42 as a balanced point for comparison to achieve higher correlation coefficients without losing too many data points. The comparative results indicated that HR and time-domain HRV measures obtained from r-PPG closely corresponded with those derived from ECG, as illustrated in Fig. 2. Specifically, the correlation coefficient for average HR was 0.86, 0.32 for SDNN, 0.24 for rMSSD, and 0.25 for pNN50—all achieving statistical significance ($p < 0.001$). The effect sizes were small for the rMSSD, pNN50, and ln(LF), medium for SDNN and large for HR

**Table 1 The definition of thermal imaging ROIs.**

| POI | Description | POI | Description |
|---|---|---|---|
| **Eyebrow (E)** | | **Lip (L)** | |
| 18 | Left side of left eyebrow | 48 | Left side of lip |
| 21 | Right side of left eyebrow | 49 | Outside of upper ip |
| 22 | Left side of right eyebrow | 50 | Right side of lip |
| 25 | Right side of right eyebrow | 51 | Outside of lower lip |
| **Forehead (F)** | | 52 | Upper lip |
| 58 | Forehead | 53 | Lower lip |
| **Nose (N)** | | **Cheek (C)** | |
| 28 | Upper part of the nose | 54* | Left cheek away from nose |
| 29 | Middle part of the nose | 55 | Left cheek closer to nose |
| 30 | Nose tip | 56 | Right cheek away from nose |
| **Nostril (S)** | | 57 | Right cheek closer to nose |
| 32 | Left nostril | **Chin area (CA)** | |
| 34 | Right nostril | 59 | Chin |
| | | **Throat area (TA)** | |
| | | 60 | Throat |

Note:
*POI 18 to 53 were adapted from *Kuzdeuov et al. (2022)* and POI 54 to 60 were defined by this study.

**Table 2 The correlation coefficients of the HRV measures generated by r-PPG and the referencing ECG.**

| MAE/HR | Correlation coefficient | | | | | | p-Value | | | | | | n |
|---|---|---|---|---|---|---|---|---|---|---|---|---|---|
| | HR | rMSSD | pNN50 | SDNN | ln(HF) | ln(LF) | HR | rMSSD | pNN50 | SDNN | ln(HF) | ln(LF) | |
| 0.3 | 0.97 | 0.49 | 0.47 | 0.6 | 0.18 | 0.11 | <0.001 | <0.001 | <0.001 | <0.001 | 0.158 | 0.404 | 61 |
| 0.32 | 0.96 | 0.43 | 0.41 | 0.47 | 0.18 | 0.08 | <0.001 | <0.001 | <0.001 | <0.001 | 0.099 | 0.462 | 87 |
| 0.34 | 0.94 | 0.39 | 0.28 | 0.55 | 0.06 | 0.14 | <0.001 | <0.001 | 0.003 | <0.001 | 0.487 | 0.135 | 118 |
| 0.36 | 0.91 | 0.26 | 0.2 | 0.43 | 0.05 | 0.14 | <0.001 | 0.001 | 0.012 | <0.001 | 0.541 | 0.083 | 160 |
| 0.38 | 0.86 | 0.26 | 0.25 | 0.35 | 0.05 | 0.14 | <0.001 | <0.001 | <0.001 | <0.001 | 0.441 | 0.051 | 199 |
| 0.4 | 0.85 | 0.25 | 0.26 | 0.32 | 0.06 | 0.15 | <0.001 | <0.001 | <0.001 | <0.001 | 0.308 | 0.02 | 251 |
| 0.42 | 0.86 | 0.24 | 0.25 | 0.32 | 0.03 | 0.14 | <0.001 | <0.001 | <0.001 | <0.001 | 0.627 | 0.017 | 285 |
| 0.44 | 0.85 | 0.21 | 0.21 | 0.32 | 0 | 0.16 | <0.001 | <0.001 | <0.001 | <0.001 | 0.938 | 0.005 | 311 |
| 0.46 | 0.84 | 0.2 | 0.19 | 0.31 | −0.01 | 0.15 | <0.001 | <0.001 | <0.001 | <0.001 | 0.863 | 0.006 | 331 |
| 0.48 | 0.84 | 0.2 | 0.19 | 0.3 | −0.01 | 0.15 | <0.001 | <0.001 | <0.001 | <0.001 | 0.877 | 0.006 | 337 |

according to *Cohen (1992)*'s criteria. However, the congruence between ln(HF) produced by r-PPG and those from the reference ECG was not statistically significant.

### Machine learning prediction

This study validated the proposed method by constructing two of the frequently used machine learning models, RF and SVM, to predict mental state changes using the facial temperature and HRV measures generated by thermal imaging and r-PPG respectively. After optimizing parameters through a grid search, the prediction accuracy for attention

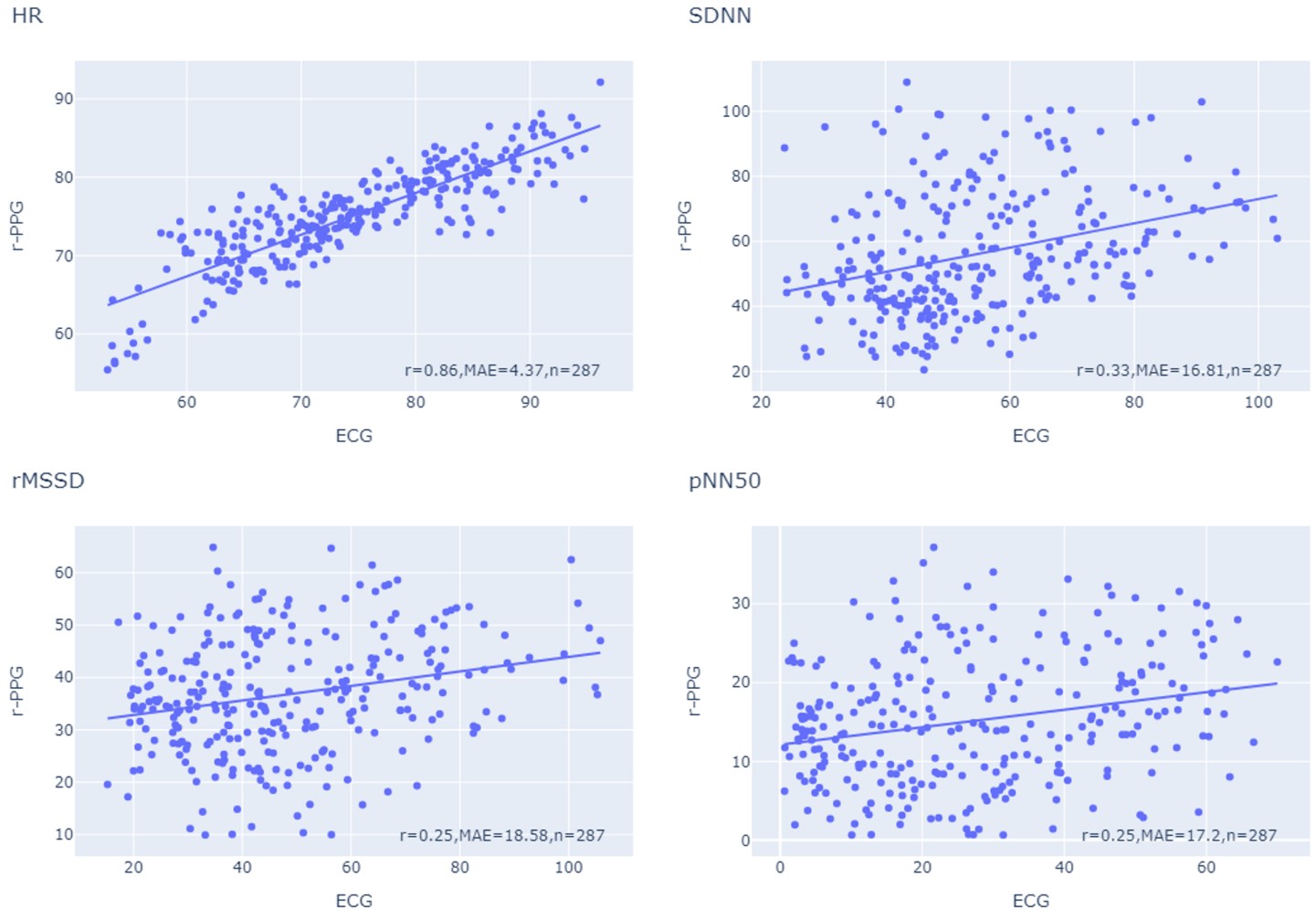

**Figure 2 Comparison of HRV measures generated by r-PPG and the reference ECG.** r, correlation coefficient; MAE, mean absolute error; *n*, number of valid data after removing outliers. MAD/HR = 0.42.

using r-PPG data with RF was 0.75 and with SVM was 0.77 (Table 3). In contrast, for moral elevation, the RF and SVM models achieved accuracies of 0.58 and 0.61, respectively, using r-PPG. Using thermal imaging data, RF and SVM models predicted cognitive stress with accuracies of 0.79 and 0.72, respectively. For moral elevation, the accuracies were 0.78 with RF and 0.75 with SVM using thermal imaging.

This study then considered two different multimodal fusion strategies to combine the data. The early fusion strategy directly employed SVM and RF models to analyze the combined features extracted by both r-PPG and thermal imaging (*Zhang et al., 2021*). This approach sought to capitalize on the inherent interdependencies between the data types by integrating them at an early stage before applying machine learning algorithms. Conversely, the late fusion strategy took a sequential approach and applied a decision tree using the Gini index to fuse the independent predictions generated by machine learning models based on two sources. This strategy banked on the strengths of individual

**Table 3 Prediction accuracy of single-modal and multimodal machine learning models.**

| Study | Mode | Model | Avg accuracy | Avg f1 |
|-------|------|-------|--------------|--------|
| Cognitive stress | rPPG | SVM | 0.77 | 0.86 |
| | | RF | 0.75 | 0.85 |
| | Thermal | SVM | 0.72 | 0.83 |
| | | RF | 0.79 | 0.87 |
| | Early fusion | SVM | 0.83 | 0.90 |
| | | RF | 0.87 | 0.91 |
| | Late fusion | Decision tree | 0.81 | 0.88 |
| Moral elevation | rPPG | SVM | 0.61 | 0.68 |
| | | RF | 0.58 | 0.63 |
| | Thermal | SVM | 0.75 | 0.79 |
| | | RF | 0.78 | 0.80 |
| | Early fusion | SVM | 0.64 | 0.70 |
| | | RF | 0.83 | 0.85 |
| | Late fusion | Decision tree | 0.75 | 0.77 |

modalities before combining them in a unified framework. In the data, the early fusion strategy outperformed the late fusion strategy and the single modal predictions with prediction accuracy of 0.87 and 0.83 using RF for cognitive stress and moral elevation respectively.

## Feature importance analysis

### Correlation analysis

The $t$-test, heatmap, and correlation coefficient analysis were frequently used tools in the feature engineering process (*Rawat & Khemchandani, 2017*). This study performed a $t$-test on the changes in the HR and HRV measures between the last 120 s to the first 120 s (Fig. 3). The results indicated that HR increased and HRV measures decreased in both the cognitive stress and moral elevation conditions. However, the differences were statistically significant only in the cognitive stress condition.

For the thermal imaging data, this study analyzed the difference between the average of the last 120 s and the average of the first 120 s. The results showed that the temperatures of the lip and cheek increased significantly when people were paying attention to the given task, and the temperatures of the nose, nostril, lip, cheek, and chin increased when the moral elevation was triggered by films (Fig. 4A). Since this study observed the temperature of different areas tended to change simultaneously, this study further investigated the relative temperature changes between different ROIs (the rows were subtracted from the column) (Figs. 4B and 4C). Since the temperature of the forehead is one of the most stable temperatures in the body (*Ioannou, Gallese & Merla, 2014*), this study followed (*Genno et al., 1997*) to choose forehead as the main comparison area. This comparison revealed that cognitive stress caused a significant relative increase only in the temperature of the lip areas. The absolute temperature change of cheek was significant but the relative

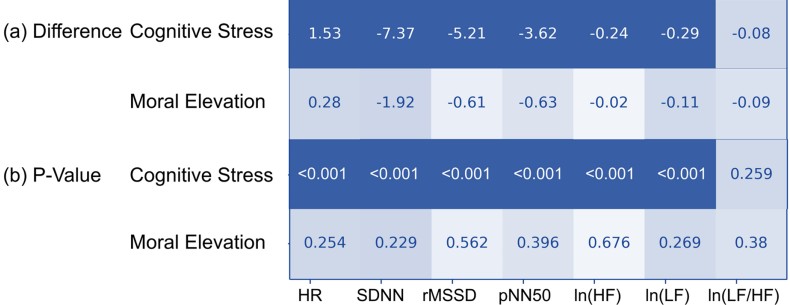

| (a) Difference | Cognitive Stress | 1.53 | -7.37 | -5.21 | -3.62 | -0.24 | -0.29 | -0.08 |
|---|---|---|---|---|---|---|---|---|
| | Moral Elevation | 0.28 | -1.92 | -0.61 | -0.63 | -0.02 | -0.11 | -0.09 |
| (b) P-Value | Cognitive Stress | <0.001 | <0.001 | <0.001 | <0.001 | <0.001 | <0.001 | 0.259 |
| | Moral Elevation | 0.254 | 0.229 | 0.562 | 0.396 | 0.676 | 0.269 | 0.38 |
| | | HR | SDNN | rMSSD | pNN50 | ln(HF) | ln(LF) | ln(LF/HF) |

**Figure 3** Heat map of HRV measurement changes induced under both cognitive stress and moral elevation conditions.

temperature changes were not. The decrease of temperature in the nose area became much more obvious, but the values did not reach statistical significance level. On the other hand, the conclusion of the relative temperature change of the moral elevation was the same as the absolute changes.

## SHAP analysis

To delve deeper into how various features impact the outcomes of black-box machine learning models, this study employed SHapley Additive exPlanations (SHAP) analysis using the Python 'shap' package (*Lundberg & Lee, 2017*). The data analysis revealed that both the RF and SVM models predominantly relied on SDNN and rMSSD features when distinguishing participants under cognitive stress caused by attention, as illustrated in Fig. 5. Additionally, the pNN50 feature emerged as a pivotal determinant in distinguishing individuals experiencing moral elevation. For thermal imaging, this study compared the top 10 features in SHAP analysis. The nasal area (ROI 28, 29, 30), the eyebrow area (ROI 18, 25), the cheeks (ROI 55, 56, 57), and the area between the nose and lip (ROI 34, 49) were essential features for thermal imaging-based mental state prediction. This study subsequently conducted SHAP analysis for the early fusion analysis. Contrary to expectations, despite thermal imaging outperforming r-PPG in single-modal prediction analysis, features generated by r-PPG dominated the early fusion analysis when variables from both modalities were combined. The important features from thermal imaging appeared to differ in the early fusion analysis compared to those in the single-modal thermal analysis.

## Linking r-PPG to thermal imaging

Generally, facial temperature was more closely related to HR than HRV in both cognitive stress and moral elevation conditions (Fig. 6). HR was negatively correlated to the temperature changes of the left eyebrow area and was positively correlated to the changes of the temperature of the cheek and outside of the lip area during the cognitive stress condition. On the other hand, HR was negatively correlated to most ROI when people were morally elevated. The HRV measures generally were less correlated to the temperature changes of the facial areas. Given that most of the correlation coefficients between facial ROIs and both HR and ln(LF)—commonly used as indicators of SNS activation—were

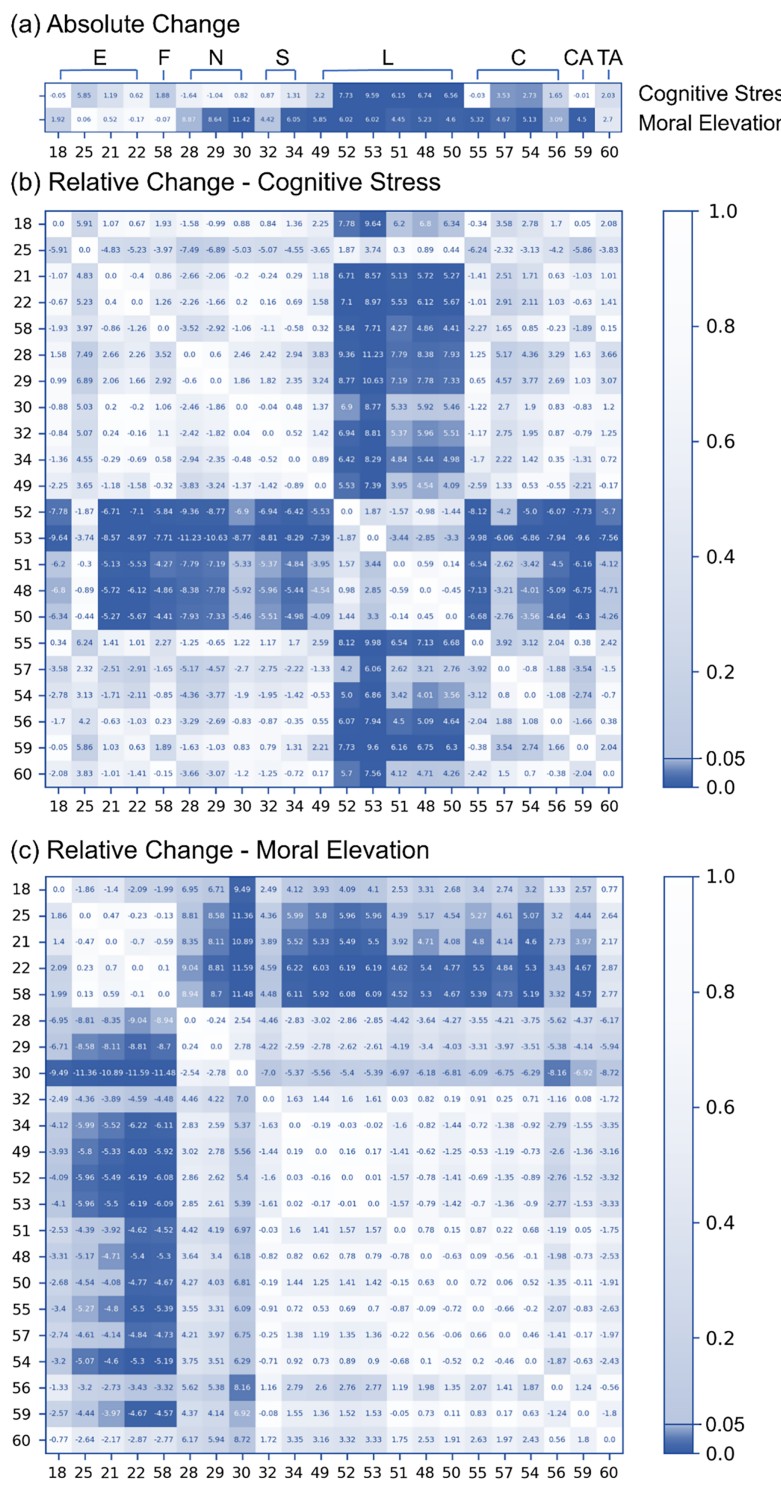

**Figure 4** Heat maps depicting changes in thermal imaging induced under both cognitive stress and moral elevation conditions.

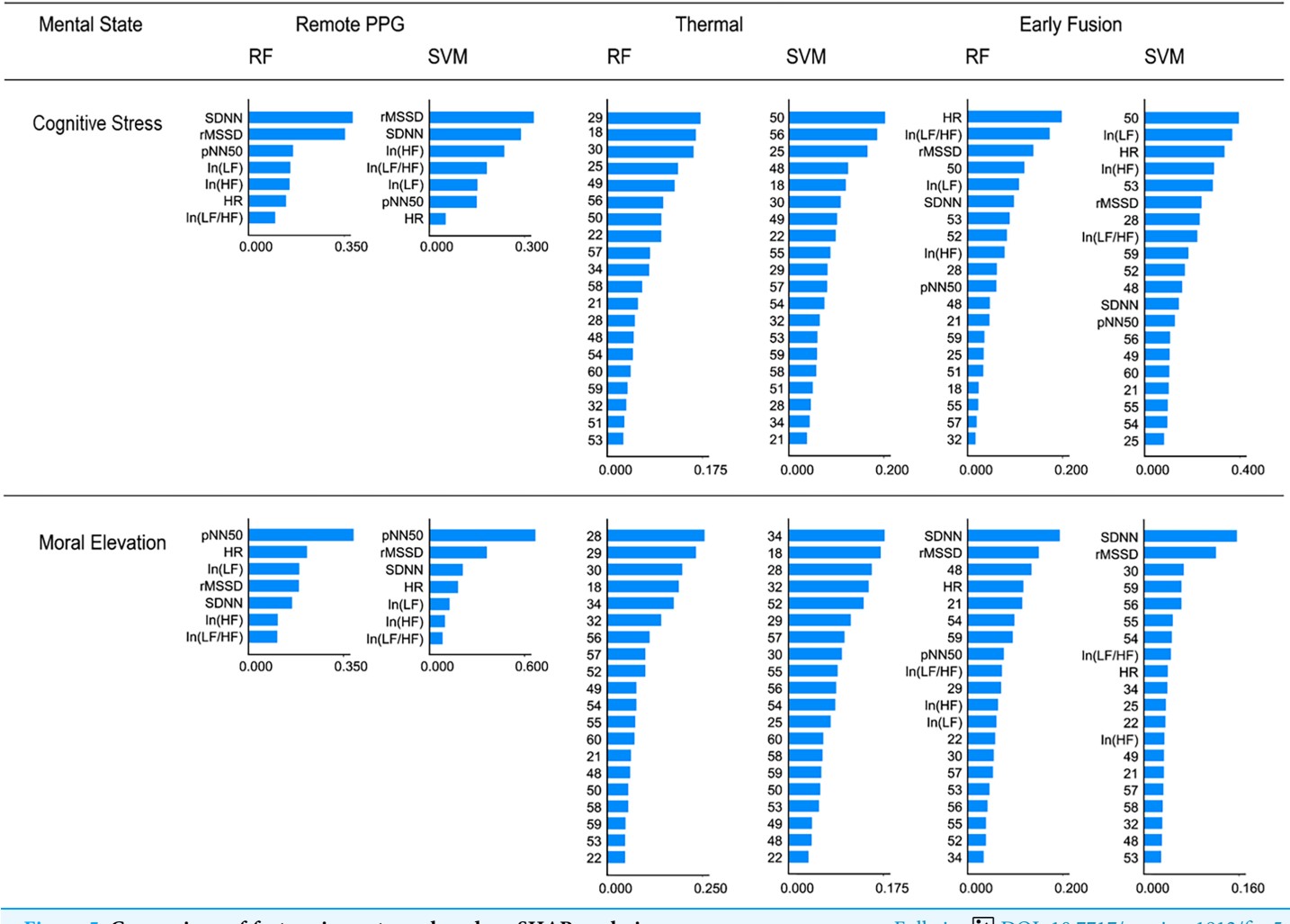

**Figure 5 Comparison of feature importance based on SHAP analysis.**               

negative, it appears that SNS activation tends to reduce facial temperature in the majority of facial areas during the moral elevation condition.

## DISCUSSION

### Principal findings

The findings of this study align with the objectives and support the use of r-PPG and thermal imaging in hidden mental state detection based on only facial skin color and temperature changes. More specifically, the results of this study can be summarized in the following aspects:

First, this study evaluated the efficacy of the multimodal approach that integrates both either r-PPG and thermal imaging to enhance prediction performance for hidden mental state changes. The accuracy of using r-PPG alone to predict mental states stood at 0.77 (SVM) for cognitive stress and 0.61 (SVM) for moral elevation. Using thermal imaging alone to predict cognitive stress and moral elevation yielded accuracies of 0.79 (RF) and

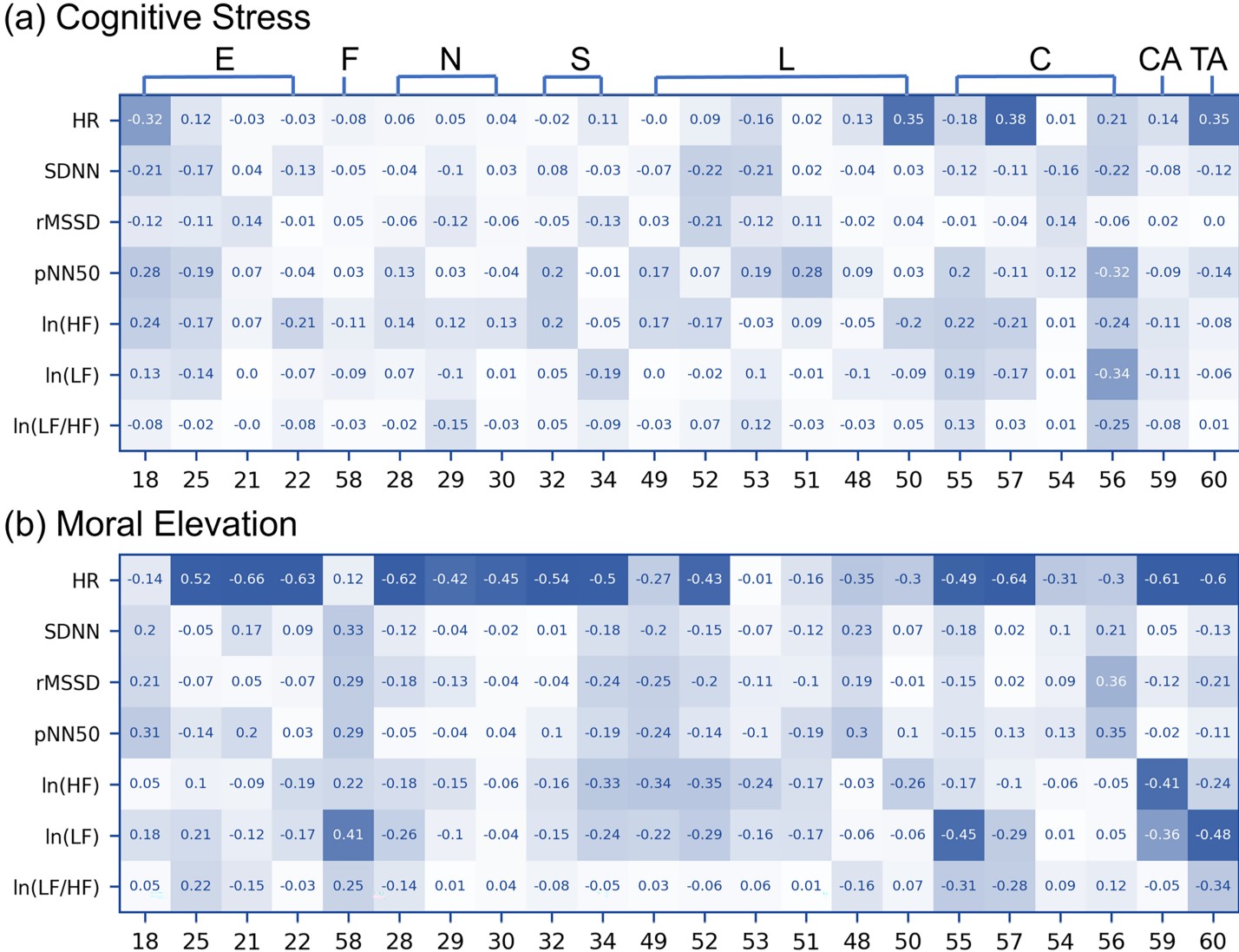

**Figure 6 Correlation coefficients between HRV measurement changes and regional facial temperature changes.** The numbers on the vertical axis represent HRV measures, while those on the horizontal axis indicate the ROI number of the face. The numbers within the boxes denote correlation coefficients. Dark blue colors signify that the *p*-values are less than 0.05 for the correlation coefficient test.

0.78 (RF), respectively. Remarkably, the early fusion approach elevated these predictive accuracies to 0.87 (RF) for cognitive stress and 0.83 (RF) for moral elevation. These results echoed the results of *Cho, Julier & Bianchi-Berthouze (2019)*, which indicated predictive accuracies for cognitive stress at 68.53% with contact PPG alone, 58.82% using only thermal imaging, and 78.33% when combining both modalities. Furthermore, compared to late fusion strategies, the findings of this study were consistent with findings from several preceding studies (*Gadzicki, Khamsehashari & Zetzsche, 2020*; *Gunes & Piccardi, 2005*) and demonstrated the superior performance of the early fusion method.

It is noteworthy that SVM and RF models are often used in the same study; however, it is difficult to explain why one model sometimes outperforms the other (*Statnikov, Wang & Aliferis, 2008*). *Fernández-Delgado et al. (2014)* reviewed 179 classifiers across 17 categories, concluding that RF was the top performer in their extensive dataset analysis. Contrarily, studies by *Ogutu, Piepho & Schulz-Streeck (2011)* and *Wainberg, Alipanahi & Frey (2016)* observed superior performances from SVM. *Boateng, Otoo & Abaye (2020)* argued that RF performs better when data is scarce, while *Grinsztajn, Oyallon & Varoquaux (2022)* found that tree-like models seem to be more robust to uninformative and non-smoothing features. The debate is further complicated by the complexity of parameter optimization; RF's performance is highly sensitive to parameter selection, as found by *Statnikov, Wang & Aliferis (2008)*, whereas SVM is less sensitive.

In this study, the unimodal analysis for predicting cognitive stress using both r-PPG and thermal imaging showed accuracies ranging from 0.72 to 0.79. For moral elevation prediction, r-PPG achieved 0.58 to 0.61, while thermal imaging attained 0.75 to 0.78. This indicates comparable SVM and RF performances, though moral elevation is less identifiable through cardiovascular features. In early fusion, integrating all variables, SVM's accuracy is more influenced by lower-accuracy variables, unlike RF, which efficiently utilizes informative features. This conclusion, drawn from a single dataset, necessitates further research for comprehensive understanding.

The SHAP analysis in this study revealed that in data fusion, cardiovascular features from r-PPG models are more influential than thermal imaging features, despite the latter's superior predictive accuracy as a single modality. This highlights two key insights: first, it underscores the enhanced predictive accuracy and benefits of multimodal fusion, combining diverse data types to overcome individual modality limitations and leveraging their combined strengths for more accurate psychological state predictions. Second, it shows that integrating multiple variables, even those with minor individual impact, significantly improves model performance, emphasizing the importance of considering a broad range of features for a comprehensive and nuanced analysis, rather than focusing only on the most dominant features.

Second, this study extends the literature on the relationship between emotions and facial temperature, applying this approach for the first time to the study of moral elevation. The data from this study showed that during experiences of moral elevation, individuals showed increased temperatures in the nose, nostrils, lips, cheeks, and chin areas—these physiological responses are clearly related to the vagus nerve system of the parasympathetic nervous system (*Haidt, 2003*). The links between cardiovascular features and mental state changes, on the other hand, are less evident, aligning with *Nhan & Chau*'s *(2009)* assertion that facial thermal imaging is more significant than HR (not HRV) and respiration. The findings also underscore *Ioannou et al.*'s *(2016)* emphasis on the importance of further exploring thermal imaging for ANS analysis, which they claimed traditionally relies heavily on HRV.

Moreover, the correlation coefficient analysis in this study revealed significant temperature increases in the lips and cheeks under cognitive stress, contrasting with previous research indicating that stress typically causes a general decrease in facial

temperature, particularly at the nose tip. However, the SHAP analysis, when using the RF model, pinpointed the nose tip as the most predictive variable. This discrepancy can be attributed to two factors. First, the relationship between stress and facial temperature might be nonlinear, meaning it may not be evident in simple correlation analyses but can become apparent in more complex, tree-structured models like RF. Second, the effects of stress on skin temperature could vary based on the stressor. While many studies have induced cognitive stress through social pressure (*Vinkers et al., 2013*), *Engert et al. (2014)* found inconsistent facial temperature responses under stress caused by physical pain and social pressure. In this study, the cognitive stress, derived from sustained attention, differs from the stress induced by physical pain or social pressure. The increased temperature around the lips and cheeks echoed the finding of *Diaz-Piedra, Gomez-Milan & Di Stasi (2019)*, who observed that sustained attention influenced arousal levels, initially raising nasal temperatures. However, since *Wang et al. (2019)* did not find a significant correlation between cognitive load and facial expressions in their EEG and thermal imaging study, the link between cognitive stress and facial temperature changes is still inconclusive and warrants further analysis.

Third, this study investigated the direct relationship between HR and facial thermography. Given the impact of SNS activity on HR, HRV, and facial temperature, this study hypothesized a direct link between these two aspects. Yet, this association has seldom been directly studied. The results of this study showed no significant relationship between cardiac features and facial temperature under cognitive stress. However, during moral elevation, a notable negative correlation emerged between HR and the temperature of the eyebrows, nose, cheeks, chin, and throat. Despite the lack of significant changes in HR during moral elevation (Fig. 3) and in the temperatures of the eyebrows and certain cheek areas (Fig. 4), a distinct correlation was observed between HR and these temperature areas (Fig. 6). This suggests concealed relationships between HR and facial temperature, necessitating additional research for a more comprehensive understanding.

Last, the data corroborated prior research, establishing that r-PPG accurately generates HR and HRV for mental state detection, with correlation coefficients of HR and SDNN between r-PPG and reference ECG at 0.86 and 0.32, respectively. Prediction accuracy for cognitive stress and moral elevation was 0.77 and 0.61. Previous studies indicate HR predictions *via* r-PPG are superior to HRV, especially in time-domain measures compared to frequency-domain HRV (*Kuss et al., 2008*). This study echoes these findings, showing HR and time-domain HRV measures as more effective. Additionally, predictive accuracy for moral elevation was lower than for cognitive stress, suggesting moral elevation may invoke subtler or more complex ANS responses, posing challenges in correlating this emotion with physiological data.

This study reinforces prior findings on the link between psychological states and HR, particularly in understanding the physiological aspects of moral elevation. Echoing ECG-based (*Eisenberg et al., 1988*) and r-PPG studies (*McDuff, Gontarek & Picard, 2014*), it observed HR increases under cognitive stress and HRV decreases due to SNS activation. However, moral elevation research is less developed. *Piper, Saslow & Saturn (2015)* observed that moral elevation might activate both sympathetic and parasympathetic

systems, affecting both HF-HRV and LF-HRV, but supporting literature is limited. This study contributes by showing no significant correlation between HR, HRV, and moral elevation. This could be due to moral elevation's complex nature, often considered a bittersweet emotion (*Oliver et al., 2018*), and its interaction with HR and HRV. Previous research shows mixed results in HR and HRV responses to emotions—for instance, sadness correlates positively with HRV and negatively with HR (*Eisenberg et al., 1988*; *Goetz, Keltner & Simon-Thomas, 2010*), while other studies note reduced HRV in sadness compared to happier states (*Goetz, Keltner & Simon-Thomas, 2010*; *Shi et al., 2017*). The inconsistent results in moral elevation are thus expected. Future research should more precisely classify moral elevation induction methods to clarify its relationship with the ANS.

## LIMITATIONS

While this study achieved most of its objectives, some data lacked statistical significance, indicating areas for methodological refinement.

First, the r-PPG data showed notable noise levels. Although signal processing has advanced, its reliability in real-world applications is still debatable. In this study, the correlation coefficient between r-PPG and ECG for HR reached 0.86 while for HRV measures was found to be only between 0.25 and 0.33, and for MAE accounted more than 20% of the mean values. The agreement with referencing devices and predictive accuracy was below several previous studies. For instance, *McDuff, Gontarek & Picard (2014)* reported a correlation coefficient between r-PPG and the reference contacted HR device for HR and HF of 1.00 and 0.93, respectively. Their research also achieved a 0.85 predictive accuracy for cognitive stress when using an SVM model on the r-PPG signal. One possible explanation for the low signal quality was the participants' movement freedom during this experiment. To maintain external validity, this study allowed participant movement, complicating data collection due to r-PPG and thermal imaging's sensitivity to motion. Despite advancements, current facial recognition algorithms, primarily designed for static images, struggled with accuracy during spontaneous movements. Furthermore, following *Pavlidis & Levine (2002)*'s methodology, participants were not required to change their hairstyles, resulting in instances where hair bangs obscured thermal signals from the forehead. Additionally, the frequent use of eyeglasses led to the exclusion of periorbital thermal imaging, omitting potential insights from areas like the supraorbital muscle, as noted by *Puri et al. (2005)*. Future research should consider adjusting experimental conditions to enhance signal quality.

Additionally, the HRV measures in this study systematically underestimated HRV in comparison to the reference ECG. This could be due to the 6-s window Fourier Analysis approach to compute HR, a method which provided similar results to the average HR in the 6-s window and inherently lowers the values of the variation of the HR. Future research should weigh the balance between noise reduction and HRV deflation. Furthermore, videos were divided into 120-s segments to monitor HRV changes per experimental phase. While some studies supported ultra-short-term HRV analysis, there is no consensus in the literature (*Pecchia et al., 2018*). This less-than-five-minute measuring period might partly

explain r-PPG's reduced predictive accuracy as previously mentioned (*Laborde, Mosley & Thayer, 2017*).

Second, the thermal imaging signal processing algorithm also requires additional refinement. In the analysis, only 55 out of 90 participants provided valid thermal imaging data, mainly due to ineffective ROI identification. The current model uses histogram of oriented gradients (HOG) and SVM techniques in a two-stage ROI identification process: initially locating the face and then pinpointing specific features. However, it often misinterprets partial facial areas as complete faces in the first stage. This could be due to the limited size and diversity of its training dataset, failing to recognize a variety of face shapes or cases with obscured faces like those with bangs, glasses, or not facing the camera directly. Head movements of participants may exacerbate this issue. The lack of established quality criteria for thermal imaging makes it challenging to filter out compromised data (*Liu, Ni & Peng, 2020a*). Future studies should consider these limitations.

Third this study explored only a limited number of data fusion methods. The exploration was confined to two techniques: early fusion, which overlooks the temporal specifics of thermal imaging, and late fusion, using a straightforward voting method. In early fusion, this study employed a simple strategy of averaging the temperature of each ROI over a 2-min recording to align thermal imaging data with r-PPG results, which represented properties over several minutes. However, this approach potentially lost detailed information. Considering the myriad data fusion strategies available (*Gandhi et al., 2023*), future research could benefit from experimenting with alternative methodologies beyond the singular approach used in this study.

Finally, this study did not sufficiently address the time delay in skin temperature changes. While HR can fluctuate within seconds, *Nakayama et al. (2005)* noted it took 220–280 s for nose temperature to revert to baseline. Rodent studies indicated varying return times based on regions: the back, head, and body took around 60–75 min, while the eyes, tails, and paws took 14, 10, and 15 min respectively (*Vianna & Carrive, 2005*). Future research should consider these delays, especially when there's a minimal gap between stimulus application and data recording.

## CONCLUSIONS

R-PPG and thermal imaging are increasingly recognized as effective tools for remotely detecting mental states. They are particularly adept at identifying subtle cognitive and emotional shifts that are less obvious in facial expressions and often missed by traditional analysis techniques. This study contributes to the academic community by validating the performance of multimodal data fusion of r-PPG and thermal imaging. It also investigates important features using both statistical analysis and explainable machine learning tools, and explores the interplay between cardiac responses and facial temperature changes in response to ANS activations. The results further corroborate the findings of previous studies regarding the effectiveness of r-PPG and thermal imaging in detecting moral elevation, a relatively understudied area. Additionally, this study has developed the 'pyMMER' package, enhancing tools available to the research community. While still in its initial stages and facing certain challenges, this study highlights the considerable potential

of these methods and the importance of their ongoing refinement and optimization. However, the statistical significance of many results fell short of the expected level, highlighting the difficulty in acquiring high-quality real-world data and the challenge in ROI detection for both r-PPG and thermal imaging in more realistic settings. Future studies should compare and explore other techniques for improving prediction accuracy, including state-of-the-art machine learning models (*Lu, Han & Zhou, 2021*; *Yu et al., 2023*). This would make the proposed method more practically useful.

## LIST OF ABBREVIATIONS

| | |
|---|---|
| **ANS** | autonomic nervous system |
| **BBI** | beat-to-beat interval |
| **BVP** | blood volume pulse |
| **ECG** | electrocardiogram |
| **HF** | absolute power of the high-frequency band (0.15–0.4 Hz) |
| **HOG** | histogram of oriented gradients |
| **HR** | heart rate |
| **HRV** | heart rate variability |
| **LF** | absolute power of the low-frequency band (0.04–0.15 Hz) |
| **ln(HF)** | the log transformation of HF |
| **ln(LF)** | the log transformation of LF |
| **MAE** | mean absolute error |
| **MMER** | multimodal emotion recognition techniques |
| **pNN50** | percentage of successive NN intervals differ by more than 50 ms |
| **PNS** | parasympathetic nervous system |
| **PPG** | photoplethysmography |
| **ROI** | region of interest |
| **rMSSD** | root mean square of successive NN interval differences |
| **r-PPG** | remote photoplethysmography |
| **SDNN** | standard deviation of the avg. normal-to-normal (NN) intervals |
| **SNS** | sympathetic nervous system |
| **SHAP** | Shapley additive explanations |
| **SVM** | support vector machines |

### Funding

This work was supported by the Beijing Normal University at Zhuhai Researcher Activation Fund (Grant No. 310432101), the Shenzhen Key Laboratory of Next Generation Interactive Media Innovative Technology (Grant No. ZDSYS20210623092001004), Shenzhen R & D Sustainable Development Funding (KCXFZ20230731093600002), the Shenzhen Key Research Base of Humanities, Social Sciences for People's Well-being Benchmarking Study (Grant No. 202003) and the

Guangdong Digital Mental Health and Intelligent Generation Laboratory (Grant No. 2023WSYS010). The funders had no role in study design, data collection and analysis, decision to publish, or preparation of the manuscript.

### Grant Disclosures
The following grant information was disclosed by the authors:
Beijing Normal University at Zhuhai Researcher Activation Fund: 310432101.
Shenzhen Key Laboratory of Next Generation Interactive Media Innovative Technology: ZDSYS20210623092001004.
Shenzhen R & D Sustainable Development Funding: KCXFZ20230731093600002.
Shenzhen Key Research Base of Humanities, Social Sciences for People's Well-being Benchmarking Study: 202003.
Guangdong Digital Mental Health and Intelligent Generation Laboratory: 2023WSYS010.

### Competing Interests
The authors declare that they have no competing interests.

### Author Contributions
- Ivan Liu conceived and designed the experiments, performed the experiments, analyzed the data, performed the computation work, authored or reviewed drafts of the article, and approved the final draft.
- Fangyuan Liu performed the experiments, analyzed the data, performed the computation work, prepared figures and/or tables, authored or reviewed drafts of the article, and approved the final draft.
- Qi Zhong performed the experiments, authored or reviewed drafts of the article, and approved the final draft.
- Fei Ma analyzed the data, authored or reviewed drafts of the article, and approved the final draft.
- Shiguang Ni conceived and designed the experiments, authored or reviewed drafts of the article, conceptualizing, funding, and approved the final draft.

### Ethics
The following information was supplied relating to ethical approvals (*i.e.*, approving body and any reference numbers):
   The research protocol was approved by the ethics committee of the faculty of psychology, Beijing Normal University (Grant no: 202203070037).

### Data Availability
   The multimodal emotion recognition (MMER) python package is available at Zenodo: 8n98324n. (2024). 8n98324n/pyMMER: Release 1 (Publication). Zenodo. https://doi.org/10.5281/zenodo.10521038.

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
