# Peer review of "Your blush gives you away: detecting hidden mental states with remote photoplethysmography and thermal imaging"

_PeerJ Computer Science, doi:10.7717/peerj-cs.1912_

## Round 0.1 · original submission · Major Revisions

Dear authors, you are advised to critically respond to all comments point by point, while preparing for the response. Please address all the comments / suggestions provided by the reviewers.

**Language Note:** The review process has identified that the English language must be improved. PeerJ can provide language editing services - please contact us at copyediting@peerj.com for pricing (be sure to provide your manuscript number and title). Alternatively, you should make your own arrangements to improve the language quality and provide details in your response letter. – PeerJ Staff

Reviewer 1 ·

Basic reporting

This work presents a remote system to predict cognitive stress and moral elevation mental states. Data is obtained using remote photoplethysmography and facial thermal imaging. The use of remote monitoring increase patient comfort.
Also, I commend the authors for providing the ’mmer’ python package to the community.

The manuscript is well written and there are minor issues with English language.
Since it is a scientific manuscript, it should be written using the third person. Expressions like “we”, “our” must be replaced accordingly.
Please correct some language issues and typos such as:
Line 242, acronym BBI is not defined.
Line 367: check -> cheek

Background theory is explained in detail.

Figures are relevant. However, Figure 6 has small text, making the reading difficult.

The provided references are relevant to the topic and well-referenced on text.

Experimental design

The study protocol received ethical approval, and all participants were informed about the study's objectives and procedures. Both documents were provided but they are not translated in English, making the reading difficult.

Applied methodology is explained in detail.

The authors have conducted an extensive study focused on two mental states: cognitive stress and moral elevation. However, after reviewing the document, it remains unclear why these specific mental states were chosen over others such as desire, motivation, depression, or anxiety.

The authors employed a video and a questionnaire as stimuli, but the order in which these stimuli were presented is unclear. The abstract (lines 27 and 28) states that "Ninety participants completed a 20-minute questionnaire, which served as a stimulus for cognitive stress, and then watched a film designed to induce the emotion of moral elevation." However, in the subsection Participants and Experiment Procedures indicates that participants watched the film before completing the questionnaire. This discrepancy should be clarified.

On lines 203 and 204, the authors state that 'Out of the 90 participants, 86 provided valid remote PPG data, and 55 provided valid thermal imaging data for both sessions.' It would be beneficial to elaborate on the criteria used to validate (or invalidate) the data.

Validity of the findings

The discussion of the results and the conclusions drawn are consistent with the data obtained.

On lines 480 and 481, the authors claim that their study is the first in thermal imaging analysis to consider relative temperature change. Have you considered using normalized temperature values? Do you believe that normalizing the temperature values could enhance the models' performance?

In Table 4, the average accuracy of the early fusion model for cognitive stress is 0.83 for SVM and 0.87 for RF (similar values). However, for moral elevation, the difference in accuracy is significantly higher (SVM: 0.64, RF: 0.83). Is it possible to justify this higher increase in accuracy for moral elevation state?

·

Basic reporting

* Clear and unambiguous, professional English used throughout.

The overall text is written clearly making it very easy to understand. Though it is essential to pay attention to the use of grammar as use of tenses is not consistent, for instance, on lines 168-170, sentence is written with a future tense, for an objective accomplished in this work.

* Literature references, sufficient field background/context provided.

General comment: Literature or Introduction section provides broad overview of the research field, however critical commentary of the literature that drives this work is not well presented. The general and broader overview of the research field can be condensed to 1-2 paragraphs, to provide more emphasis on the more recent works, further structuring into psychophysiological studies (on stress and moral elevation) and computational studies (on HRV, rPPG and thermal imaging). Please find detailed comments on this section below:

Please add adequate and most recent references throughout the text. Few of the places for example are the sentences at following lines: 56-67, 63-64, 64-65, 75-76.

83-84: Studies on nasal temperature, conducted with human participants, and the findings of the same can be mentioned, as findings on non-human primates may not entirely related to human studies.
Suggested references:
• Cho, Youngjun, Simon J. Julier, and Nadia Bianchi-Berthouze. "Instant stress: detection of perceived mental stress through smartphone photoplethysmography and thermal imaging." JMIR mental health 6.4 (2019): e10140.
• Vinkers, Christiaan H., et al. "The effect of stress on core and peripheral body temperature in humans." Stress 16.5 (2013): 520-530.

102-103: Several low cost commercial devices for PPG and ECG are available as wearable devices, and thereby portable. On the contrary, equipment for thermal imaging are generally not portable.

103-104: Minor comment that automatic ROI identification is not signal processing problem, as it pertains to computer vision. There are some recent works which have addressed these challenges:
• Chu, Wei-Ta, and Yu-Hui Liu. "Thermal facial landmark detection by deep multi-task learning." 2019 IEEE 21st international workshop on multimedia signal processing (MMSP). IEEE, 2019.
• Jitesh N Joshi, , Nadia Berthouze, Youngjun Cho. "Self-adversarial Multi-scale Contrastive Learning for Semantic Segmentation of Thermal Facial Images." 33rd British Machine Vision Conference 2022, BMVC 2022, London, UK, November 21-24, 2022. BMVA Press, 2022.

113-182: Generally at the end of “introduction” section, contributions are summarized instead of mentioning the objectives. Secondly, objectives are combined with further review of the literature, breaking the flow of reading. It is recommended to rewrite this section.

139-140: Substantial studies on HRV and mental state detection, using physiological monitoring devices exists, and on the other hand with some notable recent works, substantial progress has been made in extracting HRV using rPPG. An objective/ contribution of applying HRV derived from rPPG to mental state detection lacks sufficient novelty.
• Castaldo, Rossana, et al. "Acute mental stress assessment via short term HRV analysis in healthy adults: A systematic review with meta-analysis." Biomedical Signal Processing and Control 18 (2015): 370-377.
• Lu, Hao, Hu Han, and S. Kevin Zhou. "Dual-gan: Joint bvp and noise modeling for remote physiological measurement." Proceedings of the IEEE/CVF conference on computer vision and pattern recognition. 2021.
• Yu, Zitong, et al. "Physformer++: Facial video-based physiological measurement with slowfast temporal difference transformer." International Journal of Computer Vision 131.6 (2023): 1307-1330.

142-151: An objective/ contribution of utilizing the existing ROI methods lacks sufficient novelty required for a research article.




* Professional article structure, figures, tables. Raw data shared.

• Article structure needs significant attention. In its current state, material and methods section seems to be structured as technical report or dissertation report. Introduction lacks clear definition of the research questions as well as the key contributions, while the discussion section does not sufficiently discuss how the findings of this work relate to the prior works.
• Adequate figures and tables are provided, though the raw data could not be found in the supplemental files.
• It will be helpful to describe Figure 2 in more details, as it is hard to find the exact methods used for the steps including “data cleaning”, “video segmentation” and “ROI identification”, which are shown as identical steps for thermal and RGB. At the end of pipeline, happy and sad faces are depicted which appear not related to this study involving stress and moral elevation.




* Self-contained with relevant results to hypotheses.
Results are adequately presented with detailed figures and tables.


* Formal results should include clear definitions of all terms and theorems, and detailed proofs.
It will be helpful to have a table of abbreviations, since many terms used in this paper appear not to have been defined in the text.

Experimental design

* Original primary research within Aims and Scope of the journal.
* Research question well defined, relevant & meaningful. It is stated how research fills an identified knowledge gap.

The study protocol with two conditions without counter-balancing may be insufficient to assess the efficacy of the prediction model. For each condition, i.e. cognitive stress, and moral elevation, it is recommended to have respective baseline or control condition. Furthermore, it is recommended to collect subjective measure of stress and moral elevation from the participants as experimental conditions may not uniformly elicit stress and moral elevation in each participant, limiting the effectiveness of the prediction model, when trained only based on the intended mental states and not the actual perceived mental states.

* Rigorous investigation performed to a high technical & ethical standard.

rPPG signal extraction:
There are several recent works which have shown significantly better accuracy for real-world data, which can be utilized here.
• Lu, Hao, Hu Han, and S. Kevin Zhou. "Dual-gan: Joint bvp and noise modeling for remote physiological measurement." Proceedings of the IEEE/CVF conference on computer vision and pattern recognition. 2021.
• Yu, Zitong, et al. "Physformer++: Facial video-based physiological measurement with slowfast temporal difference transformer." International Journal of Computer Vision 131.6 (2023): 1307-1330.

HRV data processing:
Time window used as a segment for processing HRV may impact the study outcome significantly. Authors may explain their choice of time window (6-seconds), while discussing some of the recommendations provided by earlier studies.
• Li, Kai, Heinz Rüdiger, and Tjalf Ziemssen. "Spectral analysis of heart rate variability: time window matters." Frontiers in Neurology 10 (2019): 545.
• Bourdillon, Nicolas, et al. "Minimal window duration for accurate HRV recording in athletes." Frontiers in neuroscience 11 (2017): 456.

* Methods described with sufficient detail & information to replicate.

• Lines 203-204: Only 55 out of 90 participants provided valid thermal imaging data for both sessions. It would help to provide a brief explanation for the issues faced in collecting thermal imaging data.
• While 207-209 mention only ECG data being collected, 257-258 additionally mentions the use of PPG sensor. Please make this consistent in the text. It will be best to add data-collection setup figure.

Validity of the findings

* Impact and novelty not assessed. Meaningful replication encouraged where rationale & benefit to literature is clearly stated.


Correlation values presented in Table-3 indicate very weak correlation between rPPG and ECG for HRV measures. These findings highlight that HRV measures obtained from rPPG may be unsuitable for assessing ANS activity, and this needs to be highlighted in the text.

Lines 327-328, and 371-373: Findings of the study seems not congruent with major studies conducted in this space. For instance, stress is known to decrease HRV, rather than increase. It would be useful to provide discussion on these findings in relation to the earlier findings in the literature. Similarly, uncommon activation of PNS and SNS is reported in prior work on moral elevation.

Refs:
• Castaldo, Rossana, et al. "Acute mental stress assessment via short term HRV analysis in healthy adults: A systematic review with meta-analysis." Biomedical Signal Processing and Control 18 (2015): 370-377.
• Piper, Walter T., Laura R. Saslow, and Sarina R. Saturn. "Autonomic and prefrontal events during moral elevation." Biological psychology 108 (2015): 51-55.

* All underlying data have been provided; they are robust, statistically sound, & controlled.
Please refer to previous comments


* Conclusions are well stated, linked to original research question & limited to supporting results.

Authors conclude with a mention that rPPG and thermal imaging is more promising tool than facial analysis expression techniques, however there seems some missing results here as the results does not show comparison of the performance of existing facial expression analysis techniques.

Conclusion also states that this work pioneered advancements in signal processing, feature engineering and multimodal fusion, while this work uses the existing methods and does not introduce novel methods in support of this claim.

Significant portion of the text in the discussion section highlights the limitations of the research field and not the limitations of this study. This is a bit out of context for the research paper, and more appropriate for the review article.

Additional comments

It is commendable that significant efforts have been made in this work, specifically data has been acquired from large pool of participants, and interesting analysis has been conducted.

The major area to pay attention is in writing, and structure of presenting the research article. In its current state, research questions are not well defined and substantial progress made both in rPPG and thermal imaging based ROI detection has been missed.

The study protocol seems not adequate as it misses to capture subjective mental states from the participants, which may be major limitation as the experimental conditions may not elicit the mental states uniformly across the participants. The key contributions comes from the findings of psychophysiological experiment, however these findings contrast number of prior works and it is recommended to discuss these findings in relation to prior work, with an explanation of the differences.

---

## Round 0.2 · accepted · Accept

Dear authors, we are pleased to verify that you meet the reviewer's valuable feedback to improve your research.

In this round, one of the previous round reviewers wasn't able to contribute.
Thank you for considering PeerJ Computer Science and submitting your work.

Reviewer 1 ·

Basic reporting

I would like to thank the author for all the efforts to address my comments and suggestions.
The document is now written in the third person, typos were corrected, and Figure 6 was enhanced to increase readability.

Experimental design

I appreciate the authors addressing my questions about the experimental design. They provided explanatory text regarding the selection of these two mental states, the sequence of the experimental procedure, and the data validation criteria.

Validity of the findings

Thank you to the authors for their responsiveness in addressing my previous questions.